# Highly Sensitive Formaldehyde Detection Using Well-Aligned Zinc Oxide Nanosheets Synthesized by Chemical Bath Deposition Technique

**DOI:** 10.3390/ma12020250

**Published:** 2019-01-13

**Authors:** Eun-Bi Kim, Hyung-Kee Seo

**Affiliations:** School of Chemical Engineering, Chonbuk National University, Jeonju 54896, Korea; keb821@naver.com

**Keywords:** ZnO, nanosheet, formaldehyde, chemical sensor, FET, field effect transistor

## Abstract

Detection of formaldehyde is very important in terms of life protection, as it can cause serious injury to eyes, skin, mouth and gastrointestinal function if indirectly inhaled. Researchers are therefore putting effort into developing novel and sensitive devices. In this work, we have fabricated an electro-chemical sensor in the form of a field effect transistor (FET) to detect formaldehyde over a wide range (10 nM to 1 mM). For this, ZnO nanosheets (NS) were first synthesized by hydrothermal method with in-situ deposition on cleaned SiO_2_/Si (100) substrate. The synthesized materials were characterized for morphology and purity and surface area (31.718 m^2^/g). The developed device was tested for formaldehyde detection at room temperature that resulted in a linear (96%) and reproducible response with concentration, sensitivity value of 0.27 mA/M/cm^2^ with an error of ±2% and limit of detection (LOD) as 210 nM.

## 1. Introduction

Among the metal oxides, ZnO [1,2,3,4,5,6,7,8], SnO_2_ [9,10,11], TiO_2_ [12,13], Fe_2_O_3_ [14,15] and WO_3_ [16,17] are attractive materials due to their unique properties such as high electron mobility, fast electron transfer rate, material stability and so forth. One of them, ZnO has been widely used in many optoelectronic and sensing devices owing to its optical/electrical properties. Zinc oxide nanomaterial-based electrodes also exhibit excellent electrochemical activity against chemicals, biomolecules and gases due to their high electron transfer characteristics and photochemical stability. 

Detection of formaldehyde is very important in terms of life protection, as it can cause serious injury to eyes, skin, mouth and gastrointestinal function if indirectly inhaled [18,19,20,21,22]. 

Xing et al. used convenient solution combustion method for the synthesis of Ag-loaded ZnO and reported as hierarchically porous heterojunction nanocomposites and varied the Ag contents and used it for the detection of formaldehyde in gaseous form at 240 °C [23]. Wei et al. synthesized hollow nanofibers of SnO_2_-ZnO, for formaldehyde detection sensing properties and reported optimum performance at 260 °C down to 0.1 ppm with good selectivity and stability, rapid response-recovery time and high sensitivity [24]. In another report, Chen et al. used pure ZnO and graphene doped ZnO with different morphologies synthesized by hydrothermal process at 150 °C for formaldehyde gas sensing performance, in the range of 2 to 2000 ppm and delivered good selectivity and fast response/recovery time and at 200 °C [25]. Shi et al. used ZnO architectures in a three dimensional (3D) center-hollow form and studied to photoelectric gas-sensing that exhibited good selectivity to formaldehyde and excellent sensitivity at 365 nm light irradiation by conducting the measurement at room temperature [26]. Chung et al. published a review on formaldehyde gas sensing with sufficient literature survey and mentions that many methods based on spectrophotometric, fluorometric, piezoresistive, amperometric or conductive measurements have been proposed for detecting the concentration of formaldehyde in air. However, conventional formaldehyde measurement systems are bulky and expensive and require the services of highly-trained operators [27]. This has inspired us to explore the possibility of an electrochemical detection of formaldehyde in the form of a FET device that can deliver better performance at room temperature.

Mei et al. have detected formaldehyde in liquid form using a Fe/Pt modified glassy carbon electrode (GCE) showing a linear response in the range of 12.5 μM to 15.4 mM with a detection limit of 3.75 μM and a sensitivity of 40.18 μA mM^−1^cm^−2^ which was improved as compared to a Pt modified glassy carbon electrode without Fe [28]. D. Trivedi J and co-workers detected formaldehyde using Ni modified carbon electrode having linear response in the concentration range of 1 × 10^−5^–1 × 10^−3^ M with a sensitivity of 22.7 ± 3.8 μA/mM having limit of detection (LOD) of 6 μM [29]. Similarly, Nachaki and colleagues have prepared Ni-Pd modified GCE for electrochemical detection of aqueous formaldehyde, which exhibits a linear range between 10 mM to 1 mM with a sensitivity of 17 mA cm^−2^ and a detection limit of 5.4 mM [30]. 

In this work, we synthesized zinc oxide nanostructures using a simple hydrothermal method and fabricated a chemical sensor that detects formaldehyde. For sensor fabrication, synthesized nanostructures were deposited on a Si/SiO_2_ substrate using chemical bath deposition (CBD) that resulted into a uniformly aligned nanosheet electrodes. The electrochemical sensor was fabricated in the form of a FET device and the electrochemical characteristics were determined with various concentrations of formaldehyde (10 Nm–1 mM in 0.1 M) in phosphate buffer (PBS) to determine the sensing properties.

## 2. Experimental Details

### 2.1. ZnO NS Synthesis

In this work, ZnO NS was synthesized by hydrothermal synthesis using zinc nitrate hexahydrate (Zn(NO_3_)_2_·6H_2_O, ≥99.0%, Sigma Aldrich, St. Louis, MO, USA) and Urea (NH_2_CONH_2_, ≥99%, Sigma Aldrich, St. Louis, MO, USA). In a typical reaction, 0.02 M zinc nitrate hexahydrate and ~16.67% urea was dissolved in 100 mL of distilled water and stirred well for 30 min. This solution was used for synthesis of ZnO which was loaded in the Teflon coated vessel of hydrothermal reactor. In order to deposit films of ZnO during hydrothermal synthesis, the pre-coated Si/SiO_2_ substrate were dipped in the solution and vessel was sealed [31]. 

Before loading the substrate into the hydrothermal reactor, cleaned substrates were coated with a thin layer of silver through thermal evaporator that can be used as one of the electrodes of FET. The hydrothermal reaction was then carried out at 80 °C for 5 h. After cooling the reactor to room temperature, the substrates were removed and thoroughly washed with distilled water, ethanol and acetone to remove impurities and unreacted reactants. The substrate on which ZnO NS was deposited was dried in an oven at 60 °C for 12 h and then sintered at 200 °C for 2 h.

### 2.2. Material Characterization

The synthesized ZnO NS were characterized for morphology by Field Emission Scanning Electron Microscopy (FESEM, Hitachi S-4700, Tokyo, Japan) and Transmission Electron Microscopy (TEM, JEM-ARM200F, JEOL, Tokyo, Japan). Elemental analysis was done by Energy Dispersive Spectroscopy (EDS, Hitachi, Tokyo, Japan), X-ray diffractometer (XRD, Ultima IV, Rigaku, Tokyo, Japan) and Fourier Transform Infrared Spectroscopy (FTIR, Nicolet, IR 300, ThermoFisher, Waltham, MA, USA), other than the analysis of the crystallinity and component properties. The optical properties were studied by obtaining Ultraviolet-visible spectra (UV-vis, JASCO, V-670, Easton, MD, USA). Specific surface area analysis was performed to investigate the specific surface area of nanostructures using the Brunauer-Emmett-Teller (BET) Technique (Micromeritics, ASAP 2010, Norcross, GA).

### 2.3. ZnO NS FET Sensor Fabrication and Detection of Formaldehyde

For the fabrication of ZnO NS FET-sensor, p-type Si wafer with (100) orientation was cleaned by acetone, ethanol and DI water, followed by drying with nitrogen (N_2_) gas. For source and drain electrode, silver (Ag) was deposited to a thickness of ∼100–150 nm by thermal evaporation mounted with a thickness monitor. The ZnO NS layer was deposited using CBD method for 5 h at 80 °C and annealed at 200 °C for 2 h, with an expected thickness of about 20–30 µm. Zn NS was applied as channeling materials in between source and drain of the FET. In last step, the deposited Zn NS over Au-Si/SiO_2_ based FET was used for the detection of formaldehyde using the reported method [28]. 

For electrochemical detection experiments to be performed, various concentrations of formaldehyde was prepared in the range of 10 nM–1 mM in phosphate buffer solution (PBS, 0.1 M) with Ag/AgCl electrode as a counter electrode. The gate voltage (V_G_) was varied from 0 to 2 V and the resulting drain current (I_D_) value was recorded. The sensitivity was then calculated through the drain current value. The wiring diagram with electrode configuration of measurement setup is shown as Figure 1.

## 3. Result and Discussion

FE-SEM was used to analyze surface morphology and size of ZnO NS, which also gave an idea about the uniform coating of ZnO film. The micrographs in Figure 2 indicates that the ZnO NS are of sheet like structure and uniformly deposited on the substrate (Figure 2a,b, low magnification images). The thickness estimated from the image is 6 to 8 μm. The insets show the larger area view where the uniform layer and thickness is seen. Figure 2c shows the high resolution image and the element mapping (inset) obtained with EDS to investigate the composition of ZnO NS, where Zn and O elements are seen uniformly distributed and no other impurities/elements are noticed. The ZnO NS was also observed by using transmission electron microscope (TEM), high resolution transmission electron microscope (HR-TEM) and selected area diffraction pattern (SAED). Low- magnification TEM image of a ZnO NS (Figure 2d) shows that a uniform lattice without any disorders indicating pure material quality. Further, high-resolution TEM, SAED and Fast Fourier Transform (FFT) was used to confirm the lattice characteristics, which shows that the lattice grew at 0.27 nm intervals on the (011) plane. 

To investigate the specific surface area of hydrothermally synthesized ZnO NS, a specific surface area analysis was carried out, that was measured by using physical adsorption and chemical adsorption of nitrogen gas which showed a specific surface area of 31.718 m^2^/g, which is fairly large specific surface area as compared with zinc oxide of other nanostructures [32,33]. It is well know that nanostructures with larger specific surface area would have greater sensitivity owing to large surface area available for interaction/adsorption [34].

Table 1 shows specific surface area analysis of ZnO by Brunauer-Emmett-Teller (BET). The prepared sample (ZnO NSs) has the largest area value (31.718 m^2^/g) due to related large pore diameter.

Fourier transform infrared spectroscopic spectrum (FTIR, Figure 3a) shows a strong IR band at 557 cm^−1^, indicating the bonding of Zn–O, which is a metal oxide. It corresponds to the scissile vibration of water molecules and the stretching mode of O–H at 1633 cm^−1^ and 3452 cm^−1^, respectively [35]. The FTIR results showed that the synthesized material is of high purity analogous to the HRTEM results.

Phase and crystallinity of ZnO NS was confirmed by obtaining an X-ray diffraction pattern (XRD, Rigaku, CuKa, λ = 1.54178 Å) which is shown in Figure 3b. The diffraction peaks are observed at the Bragg angle of 31.73° (010), 34.329° (002), 36.19° (011), 47.44° (012), 56.52° (110), 62.70° (013), 66.28° (020), 67.382° (112), 68.98° (021), 72.35° (004) and 76.82° (014). Our ZnO Wurzite structure data agrees well with the Joint Committee on Powder Diffraction Standard (JCPDS) card no JCPDS PDF no 36-1451 [36]. Diffraction peaks other than the main peak of ZnO were not detected, which again confirmed a pure ZnO. It was found that the synthesized ZnO NS had a good crystallinity and mainly a (011) plane orientation. Figure 3c shows the absorbance spectra of the synthesized material acquired ultraviolet spectroscopy (UV-vis). It is seen that the peak absorption is at 362 nm. The optical band gap, as calculated from the absorption spectrum, is ~3.4 eV, which is the known optical band gap of the ZnO nanomaterial. Figure 3d shows the Photo Luminescence spectrum of ZnO NS obtained at room temperature, where the absorption peak is observed at 371 nm.

### Sensing Characteristics

A field effect transistor (FET) device was fabricated to develop and electrochemical sensing device for formaldehyde using ZnO NS as channel material (1 cm^2^) for drain and source and silver a gate electrode. The electrode configuration used for sensing is shown in Figure 4a. The gate voltage (V_G_) was varied from 0 to 2V and the resulting drain current (I_D_) value was recorded. The device parameters such sensitivity, minimum detection limit and regression coefficient were then calculated through the drain current value. A typical drain current (V_SG_–I_D_) curve as a function of gate voltage with and without formaldehyde is show in Figure 4b which shows a remarkable difference in the current response as magnitude of current without the formaldehyde is low, while the current observed after addition of the formaldehyde (10 nM) is higher. The data clearly shows the potential sensing characteristic of the synthesized material. Inspired with this observation, the I–V curve were then obtained with varying concentrations of formaldehyde (10 nM, 100 nM, 1µM, 10 µM, 100 µM and 1 mM in 0.1 M PBS) which are shown in Figure 4c. It can be seen that the current increases with the increasing concentration amount due to increased electron movement resulting out of the reduction of ZnO NS with formaldehyde. At the same time the current values of each curve have changed with concentration that can be used a measure of the indirect sensitivity value of the device for formaldehyde detection. For each concentration of formaldehyde, three sets of measurement were made to find the variation in the response and found that sensor was able to reproduce the result within ±2%, as shown in the Figure 4d. To estimate the sensitivity, the drain current value at the gate voltage of 1.5V was plotted with concentration, which is shown as Figure 4d. The slope of the curve is taken as the sensitivity per unit area of the deposited gate electrode. This curve was used to determine the sensitivity which is estimated as ~0.27 mA/M/cm^2^. It can be seen that the developed device is able to deliver a linear response to formaldehyde concentration to an extent of 96% (regression coefficient). The limit of detection value was calculated using Equation (1) and found as is 210 nM.
(1)$$\mathrm{LOD}=\frac{3.3\;\times \;\mathrm{standard}\;\mathrm{deviation}\;\mathrm{of}\;\mathrm{the}\;\mathrm{regression}\;}{\mathrm{slope}}$$

As it is observed from the sensing studies that the sensor is able to produce a detectable change in current with formaldehyde concentration that can be translated into a calibration curve (Figure 4d). In case of the proposed metal oxide based sensor, we believe that the physical adsorption phenomenon dominates the sensing mechanism. The reason for increased current with formaldehyde concentration is expected to be due to release of an electron from surface of ZnO due to reaction with pre-adsorbed oxygen creating an oxygen species on application of gate voltage. With increasing concentration, the amount of electron release increases resulting in increased current as observed in Figure 4c. The sensing mechanism is shown schematically in Figure 5. 

The performance of the developed device is compared with the reported sensors, some of which are listed in Table 2, which clearly shows that the developed sensors has better sensitivity values and lower detection limit, indicating the superiority of the developed sensor device compared to the reported aqueous/gas mode detection.

## 4. Conclusions

Zinc oxide nanosheet like structure was directly grown on pre-cleaned Si (100) substrate by hydrothermal synthesis at 80 °C. The synthesized materials structure was confirmed with FESEM and TEM observations while purity was confirmed with FTIR and HRTEM. With nanosheet like structure, we were able to get a specific surface area of ~31.718 m^2^/g that delivered reproducible response good sensitivity for formaldehyde in the form of a FET device at room temperature. The detection sensitivity of the formaldehyde is found as 0.27 mA/M/cm^2^ with an error of ±2% and the device is offering a detection limit up to 210 nM which is a fairly low value as comparted to reported value, indicating the possibility of using the developed FET as a commercial detection device.

## Figures and Tables

**Figure 1 materials-12-00250-f001:**
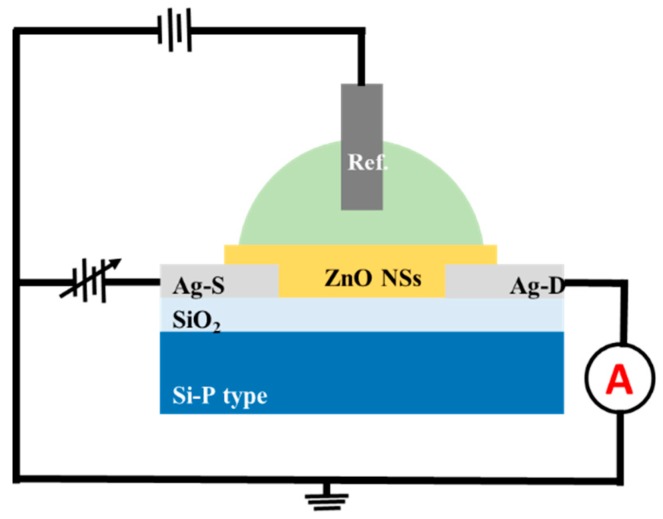
Electrode configuration and wiring diagram of the measurement setup.

**Figure 2 materials-12-00250-f002:**
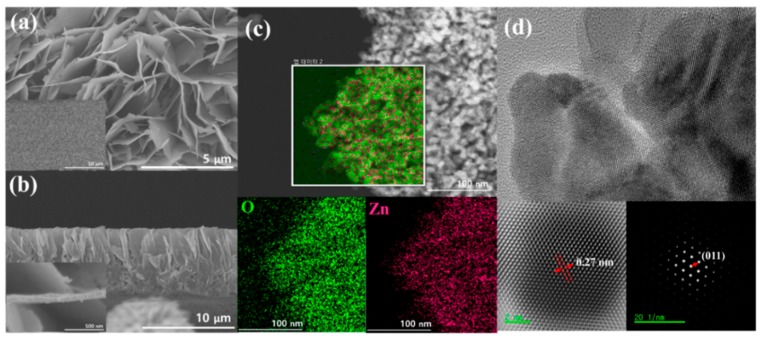
Field Emission Scanning Electron Microscopy (FESEM) image (**a**) cross section image (**b**), energy dispersive spectroscopy (EDS) mapping image (**c**) and transmission electron microscope (TEM) image (**d**) (inner HR TEM and selected area diffraction (SAED) pattern) of ZnO NS.

**Figure 3 materials-12-00250-f003:**
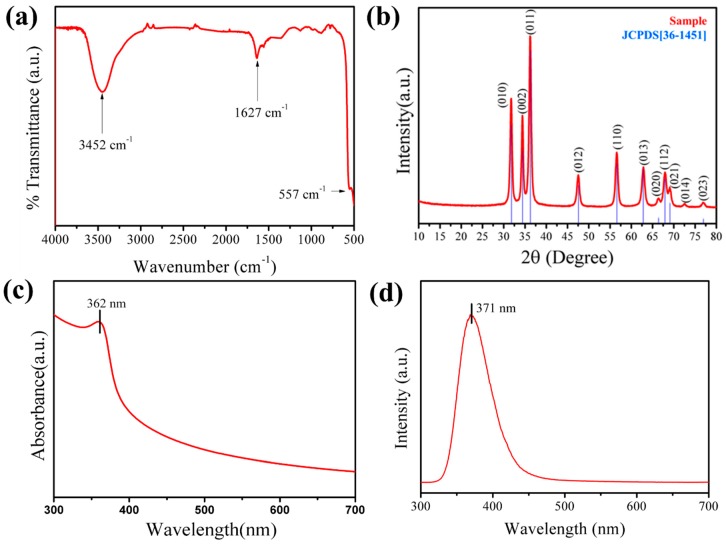
Infrared (IR) (**a**), X-ray diffraction (XRD) (**b**), UV-vis (**c**) and PL (**d**) of ZnO NS.

**Figure 4 materials-12-00250-f004:**
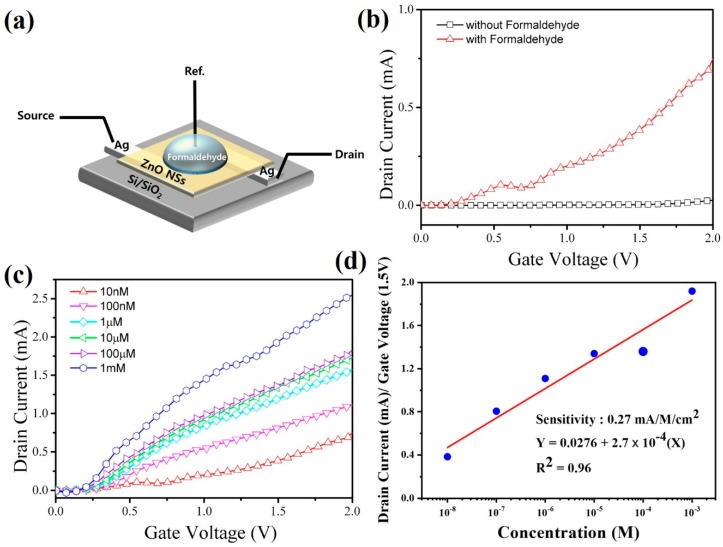
Schematic for the detection of formaldehyde (**a**), I–V curve of ZnO field effect transistor (FET) sensor in the absence and presence of formaldehyde (**b**), V_G_–I_D_ with different formaldehyde concentrations (**c**) in 0.1 M PBS solution and (**d**) calibration curve of current versus formaldehyde concentration of the fabricated FET sensor, used for sensitivity determination.

**Figure 5 materials-12-00250-f005:**
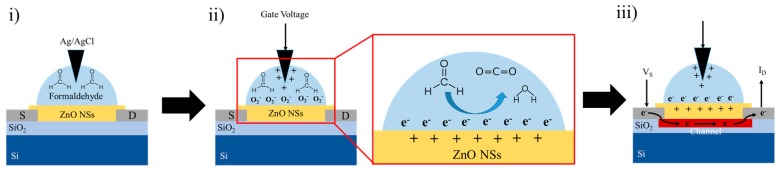
Schematic illustration of the fabricated formaldehyde sensor detection mechanism.

**Table 1 materials-12-00250-t001:** Specific surface area analysis of ZnO.

Sample	BET Surface Area (m^2^/g)	Average Pore Diameter (nm)	Reference
ZnO particle	2.3485	8.96	[32]
ZnO microflower	10.47	18.62	[33]
ZnO NSs	31.718	33.982	(present)

**Table 2 materials-12-00250-t002:** Comparison of formaldehyde sensing responses of various electrodes.

Sample	Sensitivity (mA mM^−1^ cm^−2^)	Limit of Detection	Reference
Fe/Pt/glassy carbon electrode	40.18 × 10^−3^	3.75 μM	[28] aqueous
Ni/glassy carbon electrode	22.7 ± 3.8 × 10^−3^	6 μM	[29] aqueous
Ni–Pd/GCE	17	5.4 mM	[30] aqueous
ZnO nanotubular	21.7 × 10^−^^4^	1 μM	[37]
ZnO nanoballs	4.72 × 10^−^^2^	500 μM	[38]
ZnO nanorods	105.5 × 10^−^^4^	5 nM	[39]
lotus-leaf-like ZnO	139.8 × 10^−^^4^	260 μM	[40]
ZnO NSs	2.7 × 10^−^^4^	210 nM	(present) aqueous

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
