# Peer review of "Highly Sensitive Formaldehyde Detection Using Well-Aligned Zinc Oxide Nanosheets Synthesized by Chemical Bath Deposition Technique"

_materials, 2019, doi:10.3390/ma12020250_

Reviewer 1 Report

In the work titled “Highly Sensitive Formaldehyde Detection Using Well-Aligned Zinc Oxide Nanosheets Synthesized by Chemical Bath Deposition Technique” and submitted by E.-B. Kim et al. ZnO nanosheets have been proposed as active layer in electrochemical FET sensor device to detect formaldehyde.

First of all, the authors should motivate the choice of using a sensor based on electrochemical FET; moreover, they should report the sensing mechanism.

In the introduction more references should be added; specifically, at page 1 line 23, reporting references for each cited MOx.

At page 1 line 24 substitute “;” with “,”.

In the introduction, the authors should insert a table reporting the best sensor responses of ZnO-based sensor for formaldehyde detection, reported in literature.

Please, define the acronym CBD, page 2 line51.

How have you controlled the film thickness during the deposition process? Please, discuss.

The authors should define the acronym BET, giving also briefly more information about the used procedure.

The authors should define how the sensitivity is calculated.

The authors should report the error associated to each reported measured value.

Which is the reproducibility of the process and the sensor measurements? Please, discuss.

Please, report the references for each associated FT-IR signals; moreover, the signal at 1633 cm-1 is not associated to –O-H vibration mode. Correct and report the right possible functionality to which the signal is associated. Considering the FTIR spectrum, the broad band at 3452 cm-1 associated to –O-H vibrations is probably due to the presence of not a stoichiometric ZnO; please, discuss. Therefore, how can you confirm the stoichiometric formation of ZnO? Please, discuss.

In the results and discussion section, the authors should propose a sensing mechanism.

Finally, the authors should compare the obtained sensor response towards formaldehyde detection to those reported in literature.

In my opinion the manuscript is acceptable with major revision.

The English in the manuscript is quite acceptable.

Author Response

Answer to comments:

We are thankful to the reviewers for critically evaluating the manuscript and helped us to improve the manuscript by raising questions and comments. We realized that manuscript has improved scientifically and textually with better depth and mechanism. Below we are annexing the response to reviewers’ comment point wise and hope that they will be satisfied.

 Response to Reviewer # 1

 First of all, the authors should motivate the choice of using a sensor based on electrochemical FET; moreover, they should report the sensing mechanism.

Thanks for the suggestion to add sensing mechanism. We have now added the mechanism at the end of the manuscript (page 6) depicting the reasons for change in current wit increasing concentration of formaldehyde.

 In the introduction more references should be added; specifically, at page 1 line 23, reporting references for each cited MOx.

Thanks for pointing out the references, which are added at page 1 line 22, reporting references for each cited MOx.

 At page 1 line 24 substitute “;” with “,”.

Thanks for pointing the correction, which is corrected as suggested.

 In the introduction, the authors should insert a table reporting the best sensor responses of ZnO-based sensor for formaldehyde detection, reported in literature

Thanks for the suggestion. The table is inserted at the end to present a performance comparison of the developed and reported sensor. We hope it is acceptable.

 Please, define the acronym CBD, page 2 line51.

Thanks for the suggestion. The acronym CBD has been now defined.

 How have you controlled the film thickness during the deposition process? Please, discuss.

Since it is an in-situ deposition, the thickness control is difficult and hence it was not controlled. We believe that all the sample would have similar thickness as they were synthesized in a batch.

 The authors should define the acronym BET, giving also briefly more information about the used procedure.

Thanks for the suggestion. The acronym has been defined and method is also explained in brief.

 The authors should define how the sensitivity is calculated.

The calculation of sensitivity is explained at page 5 wherein the current value at the gate voltage of 1.5V is plotted with concentration and then the slope of this curve per unit area of the gate electrode is taken as the sensitivity value.

 The authors should report the error associated to each reported measured value.

Thanks for pointing out an important parameter that was missed. For each concentration, three sets of the measurement were taken and the error was then calculated which is ±2% and is now shown in the Fig. 3d, which is used as calibration curve to determine the sensitivity.

 Which is the reproducibility of the process and the sensor measurements? Please, discuss.

The sensors were found reproducible as three sets of each measurement was taken and deviation is noted to an extent of ±2% only.

 Please, report the references for each associated FT-IR signals; moreover, the signal at 1633 cm-1 is not associated to –O-H vibration mode. Correct and report the right possible functionality to which the signal is associated. Considering the FTIR spectrum, the broad band at 3452 cm-1 associated to –O-H vibrations is probably due to the presence of not a stoichiometric ZnO; please, discuss. Therefore, how can you confirm the stoichiometric formation of ZnO? Please, discuss.

Thanks for pointing out the FTIR. It has been added a reference [32] of OH band functionality at 1633 and 3452 cm-1, respectively.

 In the results and discussion section, the authors should propose a sensing mechanism.

In the revised manuscript, a sensing mechanism is proposed and discussed.

 Finally, the authors should compare the obtained sensor response towards formaldehyde detection to those reported in literature.

Thanks for the suggestion. A table is now presented in the manuscript showing the performance comparison of the developed and reported sensor.

 In my opinion the manuscript is acceptable with major revision.

We are thankful for suggesting a major revision with useful comments, which hope fully are addressed properly.

 The English in the manuscript is quite acceptable.

Thanks for accepting the manuscript language.

Reviewer 2 Report

1. Line 28-29, it would be helpful to put some WHO references about the formaldehyde concentration that cause a specific injuries listed in 28-29 lines. 

2. Table 1 should summarized the information given in the introduction part including results obtained by the authors. I would be very helpful to compare the results, such as: operating temperature, gas-sensitive material (pure ZnO or doped-ZnO), concentration ranges. 

3. Line 51 CBD-? 

4. The measurement system photograph is needed with measurements protocol, device used for supplying the voltage and current record, etc. 

5. How many measurements were carried out at one concentration? Only 1? Fig. 3d has no error bars, the linear coefficient and curve formula should be presented as well.

Author Response

1. Line 28-29, it would be helpful to put some WHO references about the formaldehyde concentration that cause a specific injuries listed in 28-29 lines. 

As suggested, few references are added including WHO. Thanks for the suggestion.

 2. Table 1 should summarized the information given in the introduction part including results obtained by the authors. I would be very helpful to compare the results, such as: operating temperature, gas-sensitive material (pure ZnO or doped-ZnO), and concentration ranges. 

As suggested, the table has been update.

3. Line 51 CBD-? 

CBD is used as acronym for chemical bath deposition, which is now defined in the manuscript.

 4. The measurement system photograph is needed with measurements protocol, device used for supplying the voltage and current record, etc. 

As suggested, the measurement system which device used for supplying the voltage and current record is added fig. 3 and 4 in the manuscript.

 How many measurements were carried out at one concentration? Only 1? Fig. 3d has no error bars, the linear coefficient and curve formula should be presented as well.

At every concentrations, three sets of measurement was carried out and the error was to a level of ±2%, which is now shown in Fig. 3d with a linear coefficient value and equation.

Reviewer 3 Report

In this manuscript, the authors present a FET sensor for formaldehyde detection based on in-situ deposited aligned ZnO nanosheets. However, the paper seen to be not consistent. For example, in section 2.1 the authors stated to dip the substrate into the reactor for direct deposition of ZnO, while, in section 2.3, they reported the fabrication of a sensor by screen printing deposition of a ZnO powder. Again, in section 3 is discussed the characterization of a film deposited by screen printing method which is also difficult to believe observing the SEM images in Figure 1 a and b. In addition, in the introduction it is only discussed the detection of formaldehyde as gaseous form, while the manuscript is aimed to the detection of this analyte in liquid solution. The introduction should be more inherent to the work.

Other issues to be considered:

- The description of the fabricated FET device and its operation mode are not clear.

- The sensing investigation is very poor. Both selectivity and stability tests of the sensor have be performed.

- A comparison with similar reported sensors should be showed.

- Some mismatches are present in the manuscript. For example, the reported hydrothermal temperature in the conclusion. 

- The method of calculating the low detection limit must be mentioned.

- Acronyms should be mentioned at least once in the manuscript. For example, CBD in introduction section.

Author Response

In this manuscript, the authors present a FET sensor for formaldehyde detection based on in-situ deposited aligned ZnO nanosheets. However, the paper seen to be not consistent. For example, in section 2.1 the authors stated to dip the substrate into the reactor for direct deposition of ZnO, while, in section 2.3, they reported the fabrication of a sensor by screen printing deposition of a ZnO powder. Again, in section 3 is discussed the characterization of a film deposited by screen printing method which is also difficult to believe observing the SEM images in Figure 1 a and b. In addition, in the introduction it is only discussed the detection of formaldehyde as gaseous form, while the manuscript is aimed to the detection of this analyte in liquid solution. The introduction should be more inherent to the work.

Thanks for the pointing out the wrongly typed and contradictory statements regarding film preparation. In the revised manuscript it is clearly mentioned that only CBD was used for deposition.

 - The description of the fabricated FET device and its operation mode are not clear.

Thanks for pointing out missing part. The fabrication/operation details are now listed clearly.

 - The sensing investigation is very poor. Both selectivity and stability tests of the sensor have be performed.

Since the proposed work is to ascertain the feasibility of the developed device to detect formaldehyde electrochemically using a FET structure and hence only basic studies were conducted.

 - A comparison with similar reported sensors should be showed.

A table has now been added in the manuscript depicting the sensor parameters reported and obtained from the present studies for performance comparison.

 - Some mismatches are present in the manuscript. For example, the reported hydrothermal temperature in the conclusion. 

Thanks for pointing out the mistake in temperature values at two different places, which is not corrected.

 - The method of calculating the low detection limit must be mentioned.

The LOD was calculated using standard equation which is now listed in the manuscript.

 - Acronyms should be mentioned at least once in the manuscript. For example, CBD in introduction section.

Apologize for not defining the acronyms, they are now defined in the revised manuscript.

Round  2

Reviewer 1 Report

Now the revised version of the manuscript is acceptable for publication.

Author Response

We are thankful for accepting the revised manuscript.

Reviewer 3 Report

Some of the main issues of the manuscript have been resolved. However, many of the critical points remain. For example:

1) The introduction is not properly congruent with the work. It should be focused on the electrochemical detection of formaldehyde in the liquid phase and not in the gas phase. Similar papers as the followings should be considered for this purpose:

- H. Mei, W. Wu, B. Yu, H. Wu, S. Wang, and Q. Xia, "Nonenzymatic electrochemical sensor based on Fe@Pt core–shell nanoparticles for hydrogen peroxide, glucose and formaldehyde," Sensors and Actuators B: Chemical, vol. 223, pp. 68-75, 2016.

- D. Trivedi, J. Crosse, J. Tanti, A. J. Cass, and K. E. Toghill, "The electrochemical determination of formaldehyde in aqueous media using nickel modified electrodes," Sensors and Actuators B: Chemical, vol. 270, pp. 298-303, 2018.

- E. O. Nachaki, P. M. Ndangili, N. M. Naumih, and E. Masika, "Nickel-Palladium-Based Electrochemical Sensor for Quantitative Detection of Formaldehyde," ChemistrySelect, vol. 3, pp. 384-392, 2018.

2) Table 2 compares some ZnO-based electrodes for the detection of substances different than formaldehyde.

3) At line 91 the reference to figure 1 is not correct.

4) At row 95 screen printed film is still mentioned.

Author Response

Thanks for the pointing out the mistakes in the manuscript, we have now revised the manuscript incorporating all the suggestions/corrections given by you.

 1)      The introduction is not properly congruent with the work. It should be focused on the electrochemical detection of formaldehyde in the liquid phase and not in the gas phase. Similar papers as the followings should be considered for this purpose.
-The suggested references for formaldehyde sensing in aqueous form are added in the list from 27-29. A paragraph is added in the introduction between the lines 49-57.

The remaining reference list is corrected accordingly and the referred reference numbers are also corrected in the text throughout.

  2)      Table 2 compares some ZnO-based electrodes for the detection of substances different than formaldehyde.
-The values of the sensing parameters are added for the aqueous detection for newly added reference

from 27-29 for comparison. The corrected reference number in the text is highlighted.

 3)      At line 91 the reference to figure 1 is not correct.

 -Thanks for the comment. An apologies for missing Fig.1.

-Text line is now changed to 100.

- Fig.1 (was missing) is now added in the text which shows the wiring diagram used for measurement. All the Figures numbers followed by addition of Fig.1 is now cored in captions as well as in running text and are highlighted. 

 4)      At row 95 screen printed film is still mentioned.

 -The word “screen printed films” are removed and highlighted. Please check the line number 117.

The added paragraph, corrected Figure numbers and reference numbers are highlighted throughout the text.

 We hope that the above responses/corrections are accepted to the esteemed reviewer to grant the acceptance.

Once again thanking you for suggestions and corrections.

 Thanking you.

Dr. Hyung Kee Seo